# Loop-Mediated Isothermal Amplification Coupled with Reverse Line Blot Hybridization for the Detection of *Pseudomonas aeruginosa*

**DOI:** 10.3390/microorganisms12112316

**Published:** 2024-11-14

**Authors:** Daniel Ferrusca Bernal, Juan Mosqueda, Gilberto Pérez-Sánchez, José Antonio Cervantes Chávez, Mónica Neri Martínez, Angelina Rodríguez, Bertha Carvajal-Gamez

**Affiliations:** 1Immunology and Vaccines Laboratory, Natural Sciences College, Autonomous University of Queretaro, Queretaro 76010, Mexico; danielferrusca08@gmail.com (D.F.B.); joel.mosqueda@uaq.mx (J.M.); flowers_47@hotmail.com (M.N.M.); 2Cuerpo Académico, Salud Animal y Microbiología Ambiental, Natural Sciences College, Autonomous University of Queretaro, Queretaro 76010, Mexico; angelina@uaq.mx; 3Laboratorio de Psicoinmunología, Instituto Nacional de Psiquiatría Ramón de la Fuente Muñiz, Mexico City 14370, Mexico; gilberto.perez.sanchez@imp.edu.mx; 4Molecular Microbiology Laboratory, Autonomous University of Queretaro, Queretaro 76010, Mexico; jose.antonio.cervantes@uaq.mx; 5Proteogenomic and Molecular Diagnosis Laboratory, Natural Sciences College, Autonomous University of Queretaro, Queretaro 76010, Mexico; 6Maestría en Química Clínica Diagnóstica, Facultad de Química, Autonomous University of Queretaro, Queretaro 76010, Mexico

**Keywords:** *P. aeruginosa*, health-associated infections, ESKAPE, LAMP (loop-mediated isothermal amplification), reverse line blot hybridization (RBLH), detection techniques

## Abstract

*Pseudomonas aeruginosa* is a pathogen of critical priority importance according to the WHO. Due to its multi-resistance and expression of various virulence factors, it is the causal agent of severe healthcare-acquired infections (HAIs). Effective strategies to control infections caused by *P. aeruginosa* must include early and specific detection of the pathogen for early and timely antibiotic prescription. The need to develop a specific and reproducible diagnostic technique is urgent, which must often be more sensitive and faster than current clinical diagnostic methods. In this study, we implement and standardize the loop-mediated isothermal amplification (LAMP) technique, coupled with the reverse line blot hybridization (RLBH) technique for the detection of *P. aeruginosa*. A set of primers and probes was designed to amplify a specific region of the *P. aeruginosa* 16s rRNA gene. The sensitivity of the LAMP-RLBH method was 3 × 10^−4^ ng/μL, 1000 times more sensitive than the PCR and LAMP technique (this work), with a sensitivity of 3 × 10^−3^ ng/μL. The LAMP-RLBH and LAMP techniques showed specific amplification and no cross-reaction with members of the ESKAPE group and other *Pseudomonas* species. The present investigation provides a technique that can be easily performed in less time, achieving a faster and more reliable alternative compared to traditional microbial diagnostic methods for the detection of *P. aeruginosa*.

## 1. Introduction

*Pseudomonas aeruginosa* is a Gram-negative bacterium included in the priority list of 41 medical care since it belongs to the family of pathogenic bacteria for human health. It causes opportunistic infections in immunocompromised and hospitalized patients. Its wide range of antibiotic resistance and adaptability has allowed it to survive in diverse natural and artificial environments [1,2,3,4]. *P. aeruginosa* belongs to the group identified by the WHO with the acronym ESKAPE (*Enterococcus faecium*, *Staphylococcus aureus*, *Klebsiella pneumoniae*, *Acinetobacter baumannii*, *P. aeruginosa,* and *Enterobacter* sp.). They are characterized by their high survival on surfaces and medical devices due to their ability to form biofilms, which is associated with their high degree of resistance to antibiotics, acting as a physical barrier and regulating genes related to virulence and antimicrobial resistance, and being the cause of healthcare-acquired infections (HAIs) [4,5,6].

In Mexico, the Hospital Network for Epidemiological Surveillance (RHOVE) reported 61,969 HAIs in 2017, and 13,975 in 2023. The primary pathogens identified were *E. coli* (18.8%), *P. aeruginosa* (12.4%), and *K. pneumoniae* (9.5%). Additionally, *P. aeruginosa* ranks first among the microorganisms identified in the medical devices of adult and pediatric intensive care units. In addition, it is the most frequent etiological agent of bloodstream infection, pneumonia, urinary tract infections, surgical site infections, and skin and soft tissue infections associated with healthcare. This pathogen represents Mexico’s second most common cause of outbreaks and is responsible for 10 to 15% of worldwide nosocomial infections [7].

This healthcare problem has worsened due to the health emergency caused by the emergence of severe acute respiratory syndrome coronavirus 2 (SARS-CoV-2). Retrospective studies have shown that *P. aeruginosa* is present in ~10% of patients with COVID-19 admitted into the intensive care unit [8,9]. Furthermore, the mortality of HAIs that these bacteria can cause exceeds 50% worldwide, given the broad drug resistance [10]. Currently, the detection of clinically relevant bacteria, such as *P. aeruginosa*, is by isolation of the pathogenic bacteria and subsequent detection by biochemical tests, processes performed by automated equipment, allowing microbial identification, which also processes detection of antimicrobial resistance [11,12]. Unfortunately, these processes require a long time for result delivery.

The development of alternative tests to those based on isolation is being sought, which must be fast, specific, sensitive, and less expensive than conventional techniques. The LAMP (loop-mediated isothermal amplification) technique is a simple, fast, specific, sensitive, and inexpensive method for isothermal amplification of nucleic acids since it does not require specialized equipment compared to conventional polymerase chain reaction (PCR), and the reaction can be carried out in a thermoblock or water bath [13,14]. The LAMP technique has been used in other studies for the detection of *P. aeruginosa* through the amplification of *OprL*, *cytochrome b*, *ecfX*, *exoS*, and *exoU* genes [15,16,17,18].

Moreover, the LAMP technique has been combined with other molecular techniques, which increase parameters such as sensitivity and specificity, such as RT-LAMP, QRT-LAMP, LAMP-Multiplex, LAMP-lateral flow dipstick (LAMP-LFD), and LAMP-reverse line blot hybridization (LAMP-RBLH) [19,20,21,22]. The RBLH technique is a probe hybridization technique [23]. It can be conjugated with other amplification techniques such as PCR, RT-PCR, PCR-Multiplex, Nester-PCR, and RT-LAMP, which can increase the sensitivity and specificity of these techniques [24,25,26].

The RBLH technique and its variants have been used for the molecular detection of epidemiologically relevant microorganisms such as *Mycobacterium tuberculosis*, *P. aeruginosa*, *Salmonella* spp., influenza viruses, and the detection of viruses of animal health importance, such as porcine epidemic diarrhea virus and Taura syndrome virus (TSV) [23,24,25,27,28,29,30]. In the latter, they performed TSV detection through the LAMP-RBLH technique, demonstrating that it is easier and more straightforward than conventional methods because it does not require sophisticated equipment compared to RT-Nester PCR and PCR [19]. This work aims to develop a sensitive, specific, rapid, and accessible biotin-developed LAMP-RBLH method for the detection of *P. aeruginosa*.

## 2. Materials and Methods

### 2.1. Control DNA Samples and Clinical Samples

*P. aeruginosa*, *Klebsiella pneumoniae*, *Escherichia coli*, *Staphylococcus aureus*, *Enterobacter cloacae*, *Proteus mirabilis*, and *Acinetobacter baumannii* DNA positive controls used in this study were donated by the DNA bank of the “Laboratorio de Inmunología y Vacunas (LINVAS) of the Universidad Autónoma de Querétaro (UAQ). *P. aeruginosa* ATCC27853 *P. flavescens* (NR_025947), *P. resinovorans* (NR_026534), *P. stutzeri* (AB680573), *P. putida* (MF838693), *P. oryzihabitans* (NR_025881), *P. fluorescens* (P147_MH518306), *P. chlororaphis aureofaciens* (NR_042939), *P. tolaasii* (LMG_NR_041799), and *P. marginalis* (LC409077.1), used as a reference strain, were donated by José Antonio Cervantes Chavez (Universidad Autónoma de Querétaro, Querétaro). Moreover, 20 DNA samples were extracted from bacterial cultures from: blood, urine, nasal, lower respiratory tract, secretions, catheter, cerebrospinal fluid, and dialysis, donated by the DNA bank of the “Laboratorio de Inmunología y Vacunas (LINVAS) of the Universidad Autónoma de Querétaro (UAQ). All DNA samples were obtained from bacterial cultures previously identified, using the VITEK-2 method, and confirmed by sequencing. The samples were handled under the World Health Organization laboratory biosafety manual standards [31]. DNA samples were used for polymerase chain reaction (PCR) molecular techniques and loop-mediated isothermal amplification (LAMP). The negative control for each assay consists of adding nuclease-free water.

### 2.2. Quantification and DNA Integrity of P. aeruginosa

The DNA concentrations of control DNA samples and clinical samples were analyzed by a Nanodrop2000 spectrophotometer (Thermo Scientific™, Waltham, MA, USA) using the 260 nm wavelength and 260/280 nm ratio. DNA integrity was verified by performing electrophoresis on 0.8% agarose gels, stained with GelRed (Biotium, Fremont, CA, USA), and visualized on a photodocumenter to verify integrity (Bio-Rad Laboratories, Hercules, CA, USA).

### 2.3. Oligonucleotide and Probe Design

Oligonucleotide design was performed following Notomi et al. criteria [12]. First, the reference nucleotide sequence of the *P. aeruginosa* 16s rRNA gene was identified in GenBank (NR_026078.1). Subsequently, BLAST (https://blast.ncbi.nlm.nih.gov/Blast.cgi, accessed on 25 February 2018) was performed, and all the phasic sequences of the 16s rRNA gene from the different *P. aeruginosa* strains reported to date were selected. A multiple sequence alignment was performed with the MUSCLE (http://www.ebi.ac.uk/Tools/msa/muscle/, accessed on 25 February 2018) program to identify a conserved region within the different sequences. The Forward Inner Primer (FIP) and Backward Inner Primer (BIP) oligonucleotides used in the LAMP technique were labeled at their 5′-end with a biotin for subsequent binding of the amplified product to streptavidin. The probe was designed following the recommendations of Aquino de Muro (2008) [32] by adding an amino group at its 5′-end to hybridize the probe to the nylon membrane. The designed probe is complementary to the target sequence’s FIP and BIP oligonucleotide regions. Oligonucleotides previously reported for molecular detection of *P. aeruginosa* through PCR amplification of the *ecfX* gene [17] were used as controls for the experiments. The oligonucleotides and probes used in this study are shown in Table 1. The company OligoT4 synthesized the oligonucleotides and probes used in the experiments.

### 2.4. Standardization of LAMP Technique

The reaction mix used in the LAMP technique was as follows: 5 mM external oligonucleotides F3 and B3., 40 mM internal oligonucleotides FIP and BIP., 1.4 mM dNTPs (Invitrogen, Carlsbad, CA, USA), 2 mM MgSO4 (New England Biolabs, Beverly, MA, USA), 1× buffer (New England Biolabs, Beverly, MA, USA), 8 U Bst DNA polymerase (New England Biolabs, Beverly, MA, USA), and 30 ng of *P. aeruginosa* ATCC27853 target DNA. Different amplification temperatures (60–65 °C) were evaluated for standardization of LAMP techniques for 30 min, followed by 80 °C for 3 min. The LAMP products were analyzed by electrophoresis agarose gel on a 1.2% stain with GelRed^®^ (Biotium, Hayward, CA, USA) and visualized under an ultraviolet (UV) light. The test was performed in triplicate.

### 2.5. Specificity and Sensitivity of LAMP and LAMP-RBLH Techniques

The analytical sensitivity of both molecular techniques, LAMP and LAMP-RBLH, for detecting *P. aeruginosa* was determined using a concentration of 30 ng/µL of DNA from clinical samples and controls. Serial dilutions of 10^−2^–10^−9^ were performed to determine the detection limit and reproducibility of the techniques starting from known concentrations of 30 ng/µL. Analytical specificity was determined with genomic DNA of *P. aeruginosa* and different species of clinically relevant bacteria such as *K. pneumoniae*, *E. coli*, *S. aureus*, *E. cloacae*, *P. mirabilis*, *A. baumannii*, *P. aeruginosa* ATCC 27853, *P. flavescens*, *P. resinovorans*, *P. stutzeri*, *P. putida*, *P. oryzihabitans*, *P. fluorescens*, *P. chlororaphis aureofaciens*, *P. tolaasii*, and *P. marginalis*. The LAMP reaction was analyzed by electrophoresis agarose gel on a 1.2% stain with GelRed^®^ (Biotium, Hayward, CA, USA) and visualized under an ultraviolet (UV) light.

The test was carried out three times at different periods.

### 2.6. Probe Hybridization of the Probes to the Nylon Membrane

The nylon membrane was activated with 16% EDAC (1-ethyl-3-3-3 [dimethylaminopropyl] carbodiimide) in a line pattern using the Miniblotter 45 (Immunetics, Cambridge, MA, USA) to perform the hybridization of the oligonucleotide probe on the nylon membrane (Biodyne C). This allowed for covalent binding of the oligonucleotide probes and the membrane through its amino group at the 5′-end. The probes were diluted to 100 pM using NaHCO_3_ pH 8.4 solution, and the Miniblotter slots were filled linearly. Subsequently, the membrane was washed with 2× Sodium Chloride–Sodium Phosphate–EDTA buffer (SSPE) and 0.1% SDS and incubated at 60 °C for 5 min, then placed back in the miniblotter with the lines of the oligonucleotides perpendicular to the slots of the miniblotter for subsequent filling of the perpendicular spaces with the amplified products obtained from the LAMP reaction [33].

### 2.7. Standardization of the RLBH Test

The amplified products obtained from the LAMP reaction were mixed with 2× SSPE/0.1% SDS, and then these amplified products were denatured by incubating at 90 °C for 10 min. Afterward, the membrane was incubated with 2× SSPE/0.1% SDS at 60 °C for 5 min. Subsequently, the slots were filled, and the membrane was incubated at 58 °C for 60 min to hybridize the probes with the amplified products. Finally, the membrane was washed with 2× SSPE/0.5% SDS. Two positive controls of *P. aeruginosa* ATCC 27853, and a negative control (of an unrelated probe) and control without DNA were included in each membrane.

### 2.8. Detection of Hybridization Between Oligonucleotide Probes and Amplified Products

3,3′,5,5′-Tetramethylbenzidine (TMB) (Pierce, Rockford, IL, USA) was used to develop the nylon membrane at room temperature. The RBLH membrane was incubated with 20 mL of the TMB (Thermo Fisher Scientific, Waltham, MA, USA) substrate solution at room temperature for 15 min until a color reaction was observed in the positive controls. The reaction was stopped by washing with 10 mL of distilled H_2_O. The experiment was carried out three times.

### 2.9. PCR for the Detection of P. aeruginosa Genetic Material

A molecular detection for *P. aeruginosa*, additionally to the one proposed in this work, the method reported by Lavenier et al. (2007) was carried out [34]. PCR of the *ecfX* gene was performed using the oligonucleotides shown in Table 1. The conditions for the PCR’s were: 12.5 µL of My Taq Mix ™ master mix (Bioline, London, UK), 1 µL at 10 pM of each oligonucleotide, 30 ng of DNA, and nuclease-free water. The programs used were an initial denaturation at 94 °C for 5 min (1 cycle), 35 cycles at 94 °C for 20 sec followed by 57 °C for 20 s (*ecfX*, positive control), and 58 °C for 30 s (16s ribosomal RNA, this study), 72 °C for 1 min, and a final elongation step of 72 °C for 5 min. The amplification products were analyzed on 1.2% agarose gels, stained with GelRed ^®^ (Biotium, Hayward, CA, USA), and visualized on the ChemiDoc Imaging System (Bio-Rad-USA). Experiments were performed in triplicate.

## 3. Results

### 3.1. Standardization of the LAMP Technique for the Detection of P. aeruginosa

The oligonucleotide design was performed on a conserved region of the 16s rRNA gene of *P. aeruginosa*; they amplified a 216 pb of NR_026078.1 accession number (Table 1). Moreover, to establish the optimal incubation temperature, reaction mixtures were incubated in a gradient of 60–65 °C. The result showed amplification at all temperatures tested; therefore, it was decided to incubate subsequent reactions at an intermediate temperature, 63 °C (Figure 1). Incubation time analysis showed a minimum required time of 30 min for revealing positive controls.

### 3.2. Analytical Specificity and Sensitivity of the LAMP and PCR Technique

Serial dilutions were performed starting from an initial concentration of 30 ng/µL of *P. aeruginosa* DNA to determine the sensitivity of the LAMP and PCR techniques. The limit of detection (LOD) for the LAMP technique was 3 × 10^−3^ ng/µL or 0.003 ng/µL (Figure 2). PCR amplification of both the *ecfX* (528 bp) and 16s ribosomal RNA (216 bp) genes showed a LOD of 3 × 10^−2^ ng/µL or 0.03 ng/µL (Figure 2). The same sensitivity results were observed when using clinical samples. These results indicate that the LAMP technique is a thousand times more sensitive than the PCR technique. The specificity of the technique was evaluated using DNA from *E. coli*, *S. aureus*, *K. pneumoniae*, *A. baumannii*, *P. mirabilis*, *E. cloacae*, *P. aeruginosa* ATCC 27853, *P. flavescens*, *P. resinovorans*, *P. stutzeri*, *P. putida*, *P. oryzihabitans*, *P. fluorescens*, *P. chlororaphis aureofaciens*, *P. tolaasii*, and *P. marginalis*. The DNA electrophoretic revealed only amplification products in reactions containing *P. aeruginosa* DNA, suggesting that the technique is specific for its detection (Figure 3).

### 3.3. Development of the LAMP-RBLH Technique

The analysis of the LAMP-RBLH technique was based on the absence or presence of color and its intensity once the membrane was revealed with TMB. Analysis of 19 samples from the clinical isolates indicated a positive hybridization signal using the probe specific for the 16s ribosomal gene of *P. aeruginosa*. The positive control, strain ATCC 47085, had a positive detection in all cases. Additionally, the negative controls, unrelated probe, and water presented no colorimetric signal. The analytical specificity analysis showed that both the LAMP technique and hybridization are specific for the detection of *P. aeruginosa* and do not present unspecificity when using the genetic material of *E. coli*, *S. aureus*, *K. pneumoniae*, *A. baumannii*, *P. mirabilis*, and *E. cloacae*, as confirmed by the absence of development of color in the membrane. On the other hand, the LOD was 3 × 10^−4^ ng/µL, suggesting that the combination of both techniques increases the sensitivity of the test. Additionally, the use of oligonucleotides and the probe increases the analytical specificity of this assay (Figure 4).

## 4. Discussion

*P. aeruginosa* is a relevant and priority bacterium in healthcare-acquired infections (HAIs) due to its high resistance to several antibiotics and its ability to form biofilms, which allows it to survive changing environments and evade the host immune system [1,34]. Additionally, this bacterium has several virulence factors that provide an advantage for its survival, adaptation, and invasion and play a relevant role in the pathophysiology of the infection, such as various types of proteases, endo and exotoxins, flagella, pilli, pyocyanin, hemolytic phospholipase C, and siderophores. In addition, this bacterium ranks first in resistance to carbapenems, the antibiotics of choice for the treatment of multi-resistant *P. aeruginosa* [4,35,36,37,38,39]. Despite efforts by health organizations, actions taken to counteract this resistance have been unsuccessful. From the perspective of this major global problem, healthcare personnel have been dedicated to identifying bacterial resistances using susceptibility tests based on bacterial growth; however, these require up to 48 h to show a result [40]. Currently, different molecular methods contribute to detecting the presence of bacteria in clinical samples.

In this study, a LAMP-RBLH method is proposed to detect the presence of genetic material of *P. aeruginosa* based on the amplification and hybridization of conserved sequences in the 16s ribosomal gene. Relevantly, this technique has been used in detecting other microorganisms of human health importance, such as *Mycobacterium tuberculosis*, where PCR multiplex is conjugated to RBLH for the simultaneous detection of genes related to antimicrobial resistance. Their results showed a sensitivity range of 82–97% and specificity of 97–100% with an analysis completion time of 7 h, obtaining similar results to phenotypic, resistance, and sequencing characterization [30]. In this study, the LAMP-RBLH technique was reduced in a significantly shorter time, by 1.3 h, without compromising specificity or analytical sensitivity. This technique proves to be a good alternative for detecting *P. aeruginosa*, obtaining higher sensitivity results than those of phenotypic characterizations and results similar to sequencing due to the design of oligonucleotides and the probe.

Regarding the determination of resistance, this technique allows opening one more application to determine genera, species, strains, and genetic resistance markers in a short time. On the other hand, the LAMP technique is fast, sensitive, specific, and cost-effective since it does not require specialized equipment, such as a thermal cycler or qPCR equipment [13]. Interestingly, the method proposed in this study avoids electrophoresis chambers since the reaction mixture is incubated directly on a membrane and revealed by colorimetry.

In this work, a PCR assay was performed with external primers (F3 and B3) to check their specificity, obtaining the expected band with a molecular weight of 216 bp of the 16s rRNA gene of *P. aeruginosa*. The amplification of the 16s rRNA gene performed in this work showed the same results in sensitivity and specificity as those previously reported for the *ecfX* gene of *P. aeruginosa* [41].

The specificity of the PCR, LAMP, and LAMP-RBLH assays was positive only for the reference strain of *P. aeruginosa*. This specificity is due to the fact that the oligonucleotides used in this work recognize a specific region of the 16s rRNA gene, which has a low homology compared to other species of the genus *Pseudomonas*, as well as with other unrelated pathogenic bacteria.

With the results obtained, we suggest that the LAMP-RBLH technique is an effective alternative for the specific and sensitive detection of *P. aeruginosa*, faster compared to PCR, Nested PCR, and qPCR techniques. Additionally, this technique is less expensive since the membrane with the hybridized probe can be kept refrigerated for months and used for hybridization with the LAMP product, which can be performed within a minimum of 30 min [19,23]. On the other hand, since the technique is just colorimetric, it does not require sophisticated equipment such as an electrophoresis system or photodocumenter equipment to observe the result in an agarose gel [19,23,26,27,28,29].

## Figures and Tables

**Figure 1 microorganisms-12-02316-f001:**
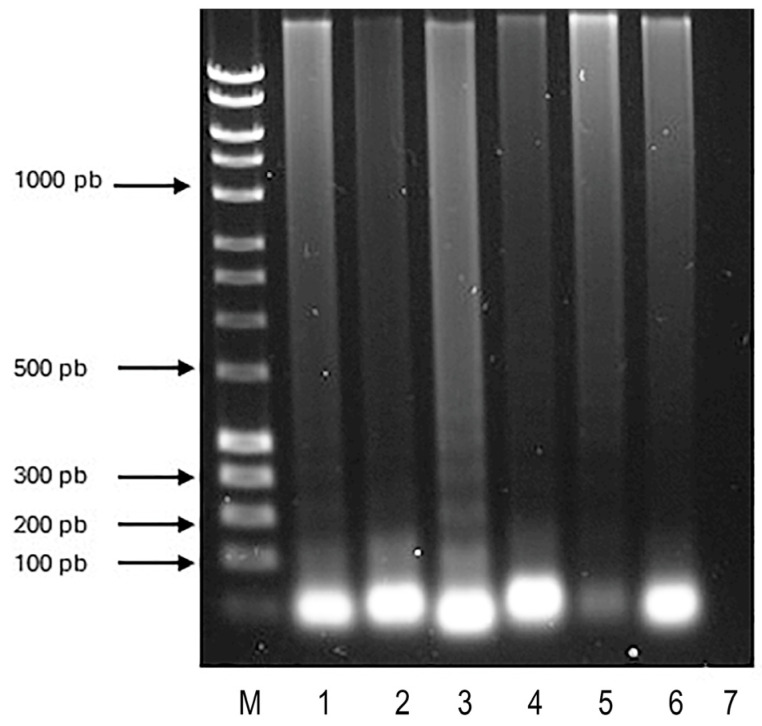
Optimization of the LAMP reaction temperature. The 1.2% electrophoresis gel stained with Red Gel ^®^ (Biotium, Hayward, CA, USA). Lane M is a 1000 bp molecular weight marker. Lane 1 is a positive control of the ATCC 27853 reference strain of *P. aeruginosa*. Lanes 2–6, LAMP reactions, incubated at different temperatures (60 °C–65 °C), *P. aeruginosa* ATCC 27853 reference strain used; Lane 7, negative control, amplified from LAMP without DNA.

**Figure 2 microorganisms-12-02316-f002:**
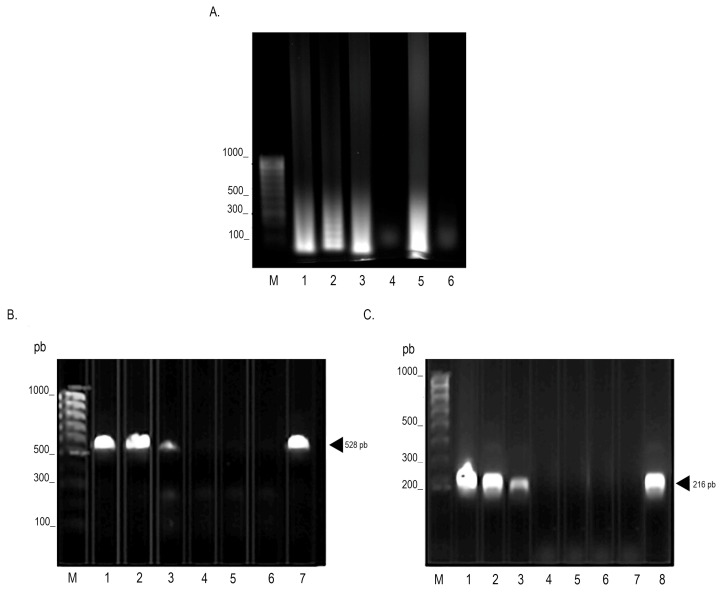
Comparison of the sensitivity of the LAMP assay and PCR using reference strain ATCC 27853 *P. aeruginosa* DNA. The 1.2% electrophoresis gel stained with Red Gel ^®^ (Biotium, Hayward, CA, USA). (**A**) Sensitivity of LAMP assay of *P. aeruginosa*. Electrophoresis gel on 1.2% staining with Red Gel ^®^ (Biotium, Hayward, CA, USA). Lane M, 1000 pb molecular weight marker; Lane 1–4, serial dilutions of the DNA amplified from the reference strain ATCC 27853 of *P. aeruginosa*. Lane 1, 3 × 10^−1^ ng/µL; Lane 2, 3 × 10^−2^ ng/µL; Lane 3, 3 × 10^−3^ ng/µL; Lane 4, 3 × 10^−4^ ng/µL; Lane 5, is a positive control of 30 ng/µL of the reference strain ATCC 27853 of *P. aeruginosa*; Lane 6, negative control, amplified from LAMP without DNA. (**B**) PCR sensitivity assay for amplification of the *ecfX* gene of *P. aeruginosa*. Lane M, 1000 pb molecular weight marker; Lane 1–4, serial dilutions of the DNA amplified from the reference strain ATCC 27853 of *P. aeruginosa*. Lane 1, 3 × 10^−1^ ng/µL; Lane 2, 3 × 10^−2^ ng/µL; Lane 3, 3 × 10^−3^ ng/µL; Lane 4, 3 × 10^−4^ ng/µL; Lanes 5 and 6 are a negative control, amplified from LAMP without DNA; Lane 7, is a positive control of 30 ng/µL of the reference strain ATCC 27853 of *P. aeruginosa*. (**C**) PCR sensitivity assay for amplification of the 16s rRNA gene of *P. aeruginosa*. Lane M, 1000 pb molecular weight marker; Lane 1–4, serial dilutions of the DNA amplified from the reference strain ATCC 27853 of *P. aeruginosa*. Lane 1, 3 × 10^−1^ ng/µL; Lane 2, 3 × 10^−2^ ng/µL; Lane 3, 3 × 10^−3^ ng/µL; Lane 4, 3 × 10^−4^ ng/µL; Lanes 6 and 7 are a negative control, amplified from LAMP without DNA; Lane 8 is a positive control of 30 ng/µL of the reference strain ATCC 27853 of *P. aeruginosa*.

**Figure 3 microorganisms-12-02316-f003:**
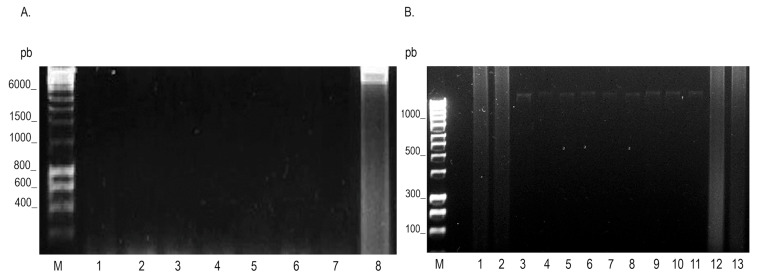
Specificity analysis of the LAMP technique for the detection of *P. aeruginosa*. The 1.2% electrophoresis gel stained with Red Gel ^®^ (Biotium, Hayward, CA, USA). (**A**) Determination of the specificity of the LAMP assay for the amplification of the 16s rRNA gene of *P. aeruginosa* with DNA ESKAPE group. Lane M, 1 kb molecular weight marker; Lane 1, negative control, amplified from LAMP without DNA; Lane 2, DNA of *K. pneumoniae*; Lane 3, DNA of *E. coli*; Lane 4, DNA of *S. aureus*; Lane 5, DNA of *E. cloacae*; Lane 6, DNA of *P. mirabilis*; Lane 7, DNA of *A. baumannii*; Lane 8, DNA of *P. aeruginosa* ATCC 27853. (**B**) Determination of the specificity of the LAMP assay for the amplification of the 16s rRNA gene of *P. aeruginosa* with the species *Pseudomonas* group. Lane M, 1000 pb molecular weight marker; Lane 1, negative control, amplified from LAMP without DNA; Lane 2, is a positive control of 30 ng/µL of the reference strain ATCC 27853 of *P. aeruginosa*; Lanes 3, 13, and 14, are DNA of *P. aeruginosa* obtained from a clinical sample; Lane 4, DNA of *P. flavescens*; Lane 5, DNA of *P. resinovorans*; Lane 6, DNA of *P. stutzeri*; Lane 7, DNA of *P. putida*; Lane 8, DNA of *P. oryzihabitans*; Lane 9, DNA of *P. fluorescens*; Lane 10, DNA of *P. chlororaphis aureofaciens*; Lane 11, DNA of *P. tolaasii*; Lane 12, DNA of *P. marginalis*.

**Figure 4 microorganisms-12-02316-f004:**
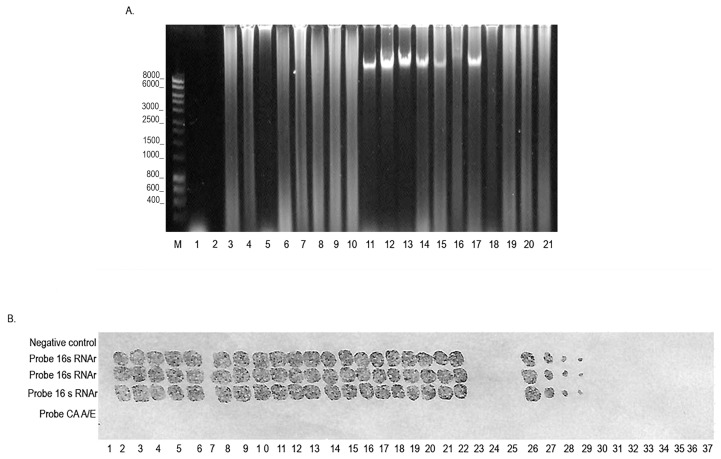
RLBH-LAMP assay for *P. aeruginosa* detection. Byodine C nylon membrane with the biotinylated *P. aeruginosa* 16s rRNA probe, revealed with Streptavidin HRP and (**A**) electrophoresis gel on 1.2% staining with Red Gel ^®^ (Biotium, Hayward, CA, USA). (**A**) LAMP assay for the detection of *P. aeruginosa* DNA obtained from cultures of clinical samples. Lane M, 1 kb molecular weight marker; Lane 1, negative control, amplified from LAMP without DNA; Lane 2, X; Lane 3–20, *P. aeruginosa* DNA obtained from cultures of clinical samples; Lane 21, is a positive control of the reference strain ATCC of *P. aeruginosa*. (**B**) Lanes 1, 7, and 37: negative control, amplified from LAMP without DNA; Lane 2: LAMP amplified from the DNA of *P. aeruginosa* strain reference ATCC; Lane 3–6 and 8–22, LAMP amplified from *P. aeruginosa* DNA clinical samples; Lane 23, LAMP with DNA of *K. pneumoniae*, as a negative control; Lane 24, LAMP with DNA of *S. aureus*, as a negative control; Lane 25, LAMP with DNA of *P. mirabilis*, as a negative control. Lanes 26–31, serial dilutions from 3 × 10^−1^ ng/µL to 3 × 10^−5^ ng/µL of DNA concentration of the reference *P. aeruginosa* strain ATCC. Lanes 32 to 36, LAMP with DNA of *E. coli* serial dilutions from 3 × 10^−1^ ng/µL to 3 × 10^−5^ ng/µL. Lane 37, without sample.

**Table 1 microorganisms-12-02316-t001:** Sequence of *P. aeruginosa*-specific LAMP primers and probes.

Primer Name	Sequence 5′-3′	Length (pb)
F3	GGTGCAAGCGTTAATCGGAATTACTGG	27
B3	CTAATCCTGTTTGCTCCCCACGC	23
FIP	/52-Bio/GGATGCAGTTCCCAGGTTGAGCCC-TTTT- GCGTAGGTGGTTCAGCAAGTTGG	51
BIP	/52-Bio/GGAAGGAACACCAGTGGCGAAGGC-TTTT- CACCTCAGTGTCAGTATCAGTCCAGG	50
F2	GCGTAGGTGGTTCAGCAAGTTGG	23
F1c	GGATGCAGTTCCCAGGTTGAGCCC	24
PROBE	/5AmMC12/GGAATTTCCTGTGTAGCGGTGAA	23

## Data Availability

The data presented in this study are available from the corresponding author upon request because studies are underway to expand the design of the method as a rapid test.

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
