# Peer review of "Loop-Mediated Isothermal Amplification Coupled with Reverse Line Blot Hybridization for the Detection of Pseudomonas aeruginosa"

_microorganisms, 2024, doi:10.3390/microorganisms12112316_

Round 1
Reviewer 1 Report
Comments and Suggestions for Authors
Here are my following comments;
1: Revise the title, it's too long, and also remove the abbreviation from the title
2:Pseudomonas aeruginosa is a Gram-negative bacterium included in the priority list of 41 medical care since belong to the family of most dangerous bacteria for human health>>>>>>>>>>>>>>Cite this ref.DOI: 10.1016/j.ijbiomac.2024.134533
3:Fig.1 looks like a table, and also it is not clear.
4: What are the advantages of combining LAMP with RBLH in terms of detection sensitivity and specificity for P. aeruginosa, and how does this hybrid approach compare to traditional methods like PCR, CFU, and sequencing in both performance and cost-effectiveness?
5: How does the choice of incubation temperature affect the efficiency and accuracy of the LAMP assay for detecting P. aeruginosa, and what is the rationale behind selecting an intermediate temperature of 63°C?
6:Given that the LAMP technique demonstrated a limit of detection (LOD) of 0.003 ng/µL, while PCR showed an LOD of 0.03 ng/µL, what specific factors contribute to the thousand-fold increase in sensitivity observed with LAMP, and how might this impact clinical diagnostic applications?
7:How does the LAMP technique ensure specificity for P. aeruginosa in the presence of closely related species and other bacterial pathogens, and what role do the designed primers and probes play in achieving this specificity?
8:How does the rapid detection capability of the LAMP-RBLH technique address the challenges of timely diagnosis and treatment of P. aeruginosa infections in clinical settings, especially in the context of antibiotic resistance?
9:Considering the elimination of electrophoresis in the LAMP-RBLH method, what are the practical implications of this change for laboratory workflows, and how does the colorimetric detection contribute to overall assay simplicity and efficiency?
Comments on the Quality of English Language
Check the grammatical errors and italics of the microbes.
Author Response
1: Revise the title, it's too long, and also remove the abbreviation from the title
Thank you for your comment. We appreciate your comment. We replacing “Loop-Mediated Isothermal Amplification (LAMP) coupled Reverse Line Blot Hybridization (RLBH) for the detection of Pseudomonas aeruginosa in clinical samples” by “Loop-Mediated Isothermal Amplification coupled Reverse Line Blot Hybridization for the detection of Pseudomonas aeruginosa ” Please see page 1, lane 2-4.
2:Pseudomonas aeruginosa is a Gram-negative bacterium included in the priority list of 41 medical care since belong to the family of most dangerous bacteria for human health>>>>>>>>>>>>>>Cite this ref.DOI: 10.1016/j.ijbiomac.2024.134533
We appreciate all your comments. We changed the introduction; please see p. 1 line 41-42, and we include a cite DOI: 10.1016/j.ijbiomac.2024.134533.
3:Fig.1 looks like a table, and also it is not clear.
We removed figure 1 from the article.
4: What are the advantages of combining LAMP with RBLH in terms of detection sensitivity and specificity for P. aeruginosa, and how does this hybrid approach compare to traditional methods like PCR, CFU, and sequencing in both performance and cost-effectiveness?
We appreciate your comment.
The LAMP technique is characterized by its speed, simplicity and high specificity and sensibility (Notomi et al., 2000, 2004). The RBLH assay is flexible, inexpensive to implement, highly sensitive, rapidity, simplicity and uses off the shelf commercially available [Gold., 2003]. The combination of both techniques (LAMP-RBLH) contributes to increasing the sensitivity and specificity of P. aeruginosa detection, which suggests reducing development times and proposing Point-of-Care Tests.
The gold standard for nucleic acid detection is polymerase chain reaction
(PCR). However, this technique requires an expensive thermocycler, trained technical personnel, and extended reaction times limits the application of PCR, and the same limitations are present in sequencing [Park et al., 2022]. Previous studies have shown that the LAMP technique is 100 to 1000 times more sensitive than the PCR technique [Fukuta et al., 2003; 2004; 2005]. The molecular techniques have made it possible to reduce the limitations of microbiological techniques such as the colony-forming unit (CFU), such as: long growth periods, culture conditions, obtaining the sample, and the experience of the staff to identify the microorganism.
Gold, B. (2003). Origin and utility of the reverse dot–blot. Expert Review of Molecular Diagnostics, 3(2), 143–152. https://doi.org/10.1586/14737159.3.2.143
Fukuta, S.; Kato, S.; Yoshida, K.; Mizukami, Y.; Ishida, A.; Ueda, J.; Michio, K.; Ishimoto, Y.
Detection of tomato yellow leaf curl virus by loop-mediated isothermal amplification reaction. J. Virol. Methods 2003, 112, 35–40.
Fukuta, S.; Ohishi, K.; Yoshida, K.; Mizukami, Y.; Ishida, A.; Kanbe, M. Development of immunocapture reverse transcription loop-mediated isothermal amplification for the detection of tomato spotted wilt virus from chrysanthemum. J. Virol. Methods 2004, 121, 49–55.
Fukuta, S.; Nimi, Y.; Ohishi, K.; Yoshimura, Y.; Anai, N.; Hotta, M.; Fukaya, M.; Kato, T.; Oya, T.; Kanbe, M. Development of reverse transcription loop-mediated isothermal amplification (RT-LAMP) method for detection of two viruses and chrysanthemum stunt viroid. Ann. Rep. Kansai Plant Prot. Soc. 2005, 47, 31–36.
Park JW. Principles and Applications of Loop-Mediated Isothermal Amplification to Point-of-Care Tests. Biosensors (Basel). 2022 Oct 10;12(10):857. doi: 10.3390/bios12100857. PMID: 36290994; PMCID: PMC9599884.
5: How does the choice of incubation temperature affect the efficiency and accuracy of the LAMP assay for detecting P. aeruginosa, and what is the rationale behind selecting an intermediate temperature of 63°C?
We appreciate your comment.
Bst polymerase operates in a temperature range of 60-65 °C, the reaction can be carried out under isothermal conditions, so the reaction is expected to be carried out in this temperature range. The choice of 63 °C does not affect the development of the technique or the result [Soroka et al., 2021]
Soroka M, Wasowicz B, Rymaszewska A. Loop-Mediated Isothermal Amplification (LAMP): The Better Sibling of PCR? Cells. 2021 Jul 29;10(8):1931. doi: 10.3390/cells10081931. PMID: 34440699; PMCID: PMC8393631.
6:Given that the LAMP technique demonstrated a limit of detection (LOD) of 0.003 ng/µL, while PCR showed an LOD of 0.03 ng/µL, what specific factors contribute to the thousand-fold increase in sensitivity observed with LAMP, and how might this impact clinical diagnostic applications?
We appreciate your comment.
The LAMP technique does not require DNA denaturation steps due to the activity of the Bst polymerase, so the reaction is carried out more quickly, with up to one billion copies being obtained in less than an hour, compared to PCR, where one million copies are obtained. In addition, the design and number of primers used in the LAMP technique, from six to eight primers, increase the sensitivity and specificity of the technique, compared to the PCR technique [Soroka et al., 2021].
7:How does the LAMP technique ensure specificity for P. aeruginosa in the presence of closely related species and other bacterial pathogens, and what role do the designed primers and probes play in achieving this specificity?
Thank you for your comment.
The specificity of the LAMP technique is based on good oligonucleotide design and analytical validation. Primers are chosen using bioinformatics analysis to determine sequence variation or conservation. Additionally, primers are optimized based on concentration, primer location, and their distance in the DNA sequence to be amplified. Primers must be single-stranded with a TM of 60-65°C and must not form stable double-stranded structures. This was done in the design of primers for P. aeruginosa and was analytically validated using DNA from the same genus and from different species.
8:How does the rapid detection capability of the LAMP-RBLH technique address the challenges of timely diagnosis and treatment of P. aeruginosa infections in clinical settings, especially in the context of antibiotic resistance?
Thank you for this observation.
Rapid, specific and sensitive detection methods provide us with additional options to microbiological or molecular methods such as PCR, which have limitations such as time and the use of sophisticated and expensive equipment. These options allow molecular techniques to be taken to laboratories that do not have specialized infrastructure such as the use of a thermocycler and provide the patient with the opportunity for adequate treatment. The LAMP-RBLH technique has advantages such as speed, sensitivity, specificity and low cost. This technique allows the inclusion of various types of probes in its design, such as: a) probes for the detection of genera and species; b) probes for the detection of genes related to antimicrobial resistance and c) probes that detect mutations.
9:Considering the elimination of electrophoresis in the LAMP-RBLH method, what are the practical implications of this change for laboratory workflows, and how does the colorimetric detection contribute to overall assay simplicity and efficiency?
We appreciate your comment.
The use of an example protocol, 30 min for the LAMP reaction, maximum 60 min of incubation of the LAMP products on the nylon membrane and maximum 15 min for the visualization of the amplicons by colorimetry, makes the analysis simple and does not require specialized equipment, which reduces the costs of the method and as an advantage it has the speed, simplicity, specificity and sensitivity of both techniques, which suggests it to be an option for the detection of microorganisms such as P. aeruginosa.

Reviewer 2 Report
Comments and Suggestions for Authors
The work aimed to provide a rapid and reliable technique for detection of Pseudomonas aeruginosa. Here are my suggestion for improving the manuscript
1) Introduction (lines 40-42): the phrase is strongly dramatic and could be improved by saying that P. aeruginosa is the main non-fermenting bacilli listed by WHO and not most dangerous. Lines later you have mentioned E. coli as prevalent, then P. aeruginosa does not be considered as dangerous. I suggest modifying the definition of P. aeruginosa.
2) Lines 47-49: please openly include the information on biofilm of ESKAPE. They also are listed because of capacity of grow as biofilms in addition to be virulent.
3) Lines 95-96: it is mandatory to italicize all microbial names, from P. aeruginosa up to Acinetobacter baumannii.
4) Lines 98-102: The same as #3, from P. aeruginosa up to P. marginalis
5) Line 165: please subscribe the number 3 in chemical formula of natrium carbonate
6) Line 178: please inform the two standard strains as ATCC XXXX and ATCC XXXX
7) Line 189: confirm if all “et al” is followed by a dot
8) Figure 1 (line 202): include the right ATCC access number used as positive control
9) Figure 2 (lines 225-226): same as #8
10) Lines 232 and 234: why does the limit of LAMP is expressed as concentration and/or mass? This is the only part of manuscript that mass is presented. The others the readers find as concentration. Please inform only the results as ng/uL
11) Figure 3 (lines 245 and 246): please put (R) correctly
12) Figure 3 (lines 247-248; 252; 2254-255; 257; and 259: it is welcome to inform the standard strains the ATTC access numbers
13) Figure 4 (lines 270; and 273-274): as same as suggestion #12
14) Line 282: inform what is the microbe which access number is ATCC 47085
15) Line 285: please italicize P.aeruiginosa
16) Figure 5 (lines 318; 320; and 324): as same as suggestion #12
17) Discussion (lines 327-330): it is welcome to rephrase the paragraph. Please include information on virulence factors (pyocyanin, biofilm…). The problem is not only antibiotic resistance.
18) Line 342: please find a new reference that include the information in the first part of this paragraph and/or rephase and make the reference 29 appears once
19) Lines 370-375: change all the information to the beginning of the paragraph. It is not your conclusions. In addition, include 2 references (first, after “…30 min”; and the second, after “…an agarose gel.”
Author Response
- Introduction (lines 40-42): the phrase is strongly dramatic and could be improved by saying that aeruginosa is the main non-fermenting bacilli listed by WHO and not most dangerous. Lines later you have mentioned E. coli as prevalent, then P. aeruginosa does not be considered as dangerous. I suggest modifying the definition of P. aeruginosa.
We appreciate all your comments. We changed the introduction; please see p. 1 line 41-42.
- Lines 47-49: please openly include the information on biofilm of ESKAPE. They also are listed because of capacity of grow as biofilms in addition to be virulent.
We appreciate your comment. We include this information; please see p. 1-2 line 46-53.
- Lines 95-96: it is mandatory to italicize all microbial names, from aeruginosa up to Acinetobacter baumannii.
All the errors were revised and corrected according; please see p. 3 line 100-101.
- Lines 98-102: The same as #3, from aeruginosa up to P. marginalis
All the errors were revised and corrected according; please see p. 3 line 103-106.
- Line 165: please subscribe the number 3 in chemical formula of natrium carbonate
We appreciate your comment. We include the number 3 in chemical formula NaHCO3; please see page 4 line 172.
- Line 178: please inform the two standard strains as ATCC XXXX and ATCC XXXX
Thank you for this observation. We include a ATCC number, please see p.4 line 185.
- Line 189: confirm if all “et al” is followed by a dot
We appreciate your comment. We include a dot after “et al.” Please see p 3 line 125, and p. 4 line 196.
- Figure 1 (line 202): include the right ATCC access number used as positive control
Thank you for this observation. We removed figure 1 from the article.
- Figure 2 (lines 225-226): same as #8
Thank you for this observation. We include a ATCC number, please see p.6 line 223-225.
- Lines 232 and 234: why does the limit of LAMP is expressed as concentration and/or mass? This is the only part of manuscript that mass is presented. The others the readers find as concentration. Please inform only the results as ng/uL
We appreciate your comment. We include ng/uL as the result. Please see p.6 line 230.
- Figure 3 (lines 245 and 246): please put (R) correctly
Thank you for this observation. We correct the mistake. Please see p. 7 line 244-245.
- Figure 3 (lines 247-248; 252; 2254-255; 257; and 259: it is welcome to inform the standard strains the ATTC access numbers
We appreciate your comment. We include this information. Please see p. 7, line 243-259.
- Figure 4 (lines 270; and 273-274): as same as suggestion #12
We include the ATCC access numbers, please see page 8 line 271, and 274.
- Line 282: inform what is the microbe which access number is ATCC 47085
Thank you for your comment. We correct the ATCC correct number, the correct number is 27853. Please see p. 8 line 284.
- Line 285: please italicize aeruginosa
Thank you for this observation. We italicize P.aeruginosa. Please see p. 8 line 287.
- Figure 5 (lines 318; 320; and 324): as same as suggestion #12
We appreciate your comment. We include this information. Please see p. 9, line 320; 321, and 326.
- Discussion (lines 327-330): it is welcome to rephrase the paragraph. Please include information on virulence factors (pyocyanin, biofilm…). The problem is not only antibiotic resistance.
We appreciate your comment. We include this information “P. aeruginosa is a relevant and priority bacterium in healthcare-acquired infections (HAIs) due to its high resistance to several antibiotics and its ability to form biofilms, which allows it to survive changing environments and evade the host immune system [34]. Additionally, this bacterium has several virulence factors that provide an advantage for its survival, adaptation, invasion and play a relevant role in the pathophysiology of the infection, such as: various types of proteases, endo and exotoxins, flagella, pilli, pyocyanin, hemolytic phospholipase C and siderophores. In addition, this bacterium ranks first in resistance to carbapenems, the antibiotics of choice for the treatment of multi-resistant P. aeruginosa [4,36,37,38,39,40]. ”, please see page 9, line 329-337.
- Line 342: please find a new reference that include the information in the first part of this paragraph and/or rephase and make the reference 29 appears once
Thank you for this observation. We include a new reference and we corrected the appearance of reference 29, please see page 9, line 337.
- Lines 370-375: change all the information to the beginning of the paragraph. It is not your conclusions. In addition, include 2 references (first, after “…30 min”; and the second, after “…an agarose gel.
We have clarified in the text, please see page 10, line 375-377:
“In this work, a PCR assay was performed with external primers (F3 and B3) to check their specificity, obtaining the expected band with a molecular weight of 216 bp, of the 16S rRNA gene of P. aeruginosa. The amplification of the 16S rRNA gene performed in this work showed the same results in sensitivity and specificity as those previously reported for the ecfX gene of P. aeruginosa [42].”.
We include a reference, please see page 10, line 363 and 367.

Round 2
Reviewer 1 Report
Comments and Suggestions for Authors
The manuscript has undergone significant revisions and is now ready for acceptance!